# Fit for What Purpose? Exploring Bicultural Frameworks for the Architectural Design of Acute Mental Health Facilities

**DOI:** 10.3390/ijerph18052343

**Published:** 2021-02-27

**Authors:** Gabrielle L. S. Jenkin, Jacqueline McIntosh, Susanna Every-Palmer

**Affiliations:** 1Department of Psychological Medicine, University of Otago, Wellington 23A Mein St., Wellington 6021, New Zealand; susanna.every-palmer@otago.ac.nz; 2School of Architecture, Victoria University of Wellington, 139 Vivian Street, Wellington 6011, New Zealand; jacqueline.mcintosh@vuw.ac.nz

**Keywords:** mental health, architecture, design, fit for purpose, recovery, bicultural

## Abstract

Acute mental health care facilities have become the modern equivalent to the old asylum, designed to provide emergency and temporary care for the acutely mentally unwell. These facilities require a model of mental health care, whether very basic or highly advanced, and an appropriately designed building facility within which to operate. Drawing on interview data from our four-year research project to examine the architectural design and social milieu of adult acute mental health wards in Aotearoa New Zealand, official documents, philosophies and models of mental health care, this paper asks what is the purpose of the adult inpatient mental health ward in a bicultural country and how can we determine the degree to which they are fit for purpose. Although we found an important lack of clarity and agreement around the purpose of the acute mental health facility, the general underpinning philosophy of mental health care in Aotearoa New Zealand was that of recovery, and the CHIME principles of recovery, with some modifications, could be translated into design principles for an architectural brief. However, further work is required to align staff, service users and official health understandings of the purpose of the acute mental health facility and the means for achieving recovery goals in a bicultural context.

## 1. Introduction

Latest WHO estimates for 2017 suggest that 792 million or 10.7% of the global population live with a mental health disorder [1]. In Aotearoa New Zealand, 46.6% of the population are predicted to experience a mental illness or an addiction at some time in their lives, with one in five people affected within any one year [2]. Specialist mental health and addictions services in Aotearoa have been experiencing year-on-year increases in demand since 2003, with a record number of 160,000 or 3.5% of the population, using these services in 2015 [3]. Approximately 15,000 of these people were admitted to an inpatient unit during 2015 [3].

The NZ government has significant investment in these services, with District Health Boards (DHBs) spending more than $200 million per annum providing inpatient mental health care. Mental health and addictions service users in these settings require the greatest levels of support, and providing care can be difficult and demanding and requires the coordination of multiple services. Failing to achieve this has enormous implications for service users, their family and whānau (wider family) and other sectors, so getting it right is a critical investment [3]. In 2017, following unprecedented electoral interest in mental health as well as widespread concerns about the high rates of suicide, the government announced a ministerial inquiry into mental health [4]. Shortly after, the government announced a significant funding investment in the rebuild and refurbishment of a number of adult acute mental health facilities [5].

Mental health services in the UK, Australia and Aotearoa all support ‘personal recovery’ as a primary orientation to their care approach, seeking to minimise levels of distress and impacts on daily living [6]. To understand recovery and to ground it in an empirical foundation for the identification and evaluation of recovery-oriented interventions, a conceptual framework was developed in 2014 by Bird et al. [7], establishing five key recovery processes: needs for connectedness, hope and optimism, identity, meaning and purpose and empowerment (CHIME). These processes are well developed in the mental health literature; however, the means for facilitating recovery remains unclear and has yet to be included as a prime architectural design consideration for the facilities themselves. Similarly, the synthesis of these processes into architectural design has not been adequately researched. As a result, architects’ limited knowledge of building user experiences of acute mental health facilities are currently based on assumptions [8], thus generating a hiatus in scientific knowledge and hampering the evidence base regarding the design of mental health care environments.

Acute mental health care in Aotearoa New Zealand is governed by the Ministry of Health, which oversees the 20 DHBs tasked with providing acute inpatient care in their regions. Excluding psychogeriatric, youth, forensic and intellectual disability facilities, there are 20 publicly funded *adult* inpatient acute mental health wards. Some of these acute mental health facilities are attached to hospitals and some exist on separate sites. Very few private or non-government acute care facilities exist. Publicly funded facilities provide short term ‘emergency’ care for people during a mental health crisis. These are not meant to be long-term residential mental health facilities as evidenced by the key performance indicator (KPI) of a 14–21 day admission target for crisis care [9]. Mental health care following discharge from acute inpatient care is provided in the community by a range of non-government organisations (NGOs) and DHB-funded professional mental health support services. These include some limited respite care and supported accommodation and professional outreach support services provided to people in their homes and communities.

Funding for health services, including infrastructure, is provided by the government but the health budget for each region is managed by the respective DHB. The architectural design of mental health facilities is contracted out to various, often local, commercial architecture firms through a competitive tendering process. The process of developing the architects’ design brief involves consultation with the DHB managerial and professional staff and more recently with service users and other key end-users (families of service users, NGOs and community groups). There also exists Ministry of Health Design Guidelines [10] and building specifications that provide some guidance on the design of acute mental health facilities and accommodation in these [11]. These are intended to support DHBs and architects as they go about the process of deciding how each facility will be designed. Once built, there is no standard process for assessing if these facilities work in the way in which they were intended or whether or not they are fit for the purpose for which they were designed.

With respect to recovery, the monitoring aspect of acute inpatient mental health care is undertaken by various bodies, including the Ministry of Health, the Health Quality and Safety Commission, District Inspectors and the Office of the Ombudsman, each of whom are tasked with reporting back to government on aspects of quality of mental health care and the facilities themselves. Standards of care, provider competence and designer competence are evaluated indirectly through the establishment of competencies, certification of the service and through architectural registration. Service user satisfaction for mental health services is measured via quantitative surveys at a very basic level. Some aspects of service provision and the environment for its delivery are assessed by the Office of the Ombudsman, reporting on the Optional Protocol to the Convention against Torture and other Cruel, Inhuman or Degrading Treatment or Punishment (OPCAT).

Models of care, whether highly advanced or very basic, require a facility within which to operate. In this paper, we examine the *purpose* of these acute mental health facilities from the perspectives of the various stakeholders in order to develop a framework that can determine whether the built facilities are ‘fit for purpose’. Drawing on official documents, stated philosophies and interviews with staff and service users, we aim to provide a holistic understanding of how the purpose is currently defined and understood, and if this might be translated into design principles for an architectural brief.

## 2. Methods

This research is part of a larger four-year study ‘Design of Acute Mental Health Wards: The New Zealand Experience’ to examine the design and social milieu of acute adult inpatient mental health wards in Aotearoa New Zealand.

### 2.1. Ethics, Consultation and Locality Approvals

Prior to funding of the research, consultation with the Ngāi Tahu Research Consultation Committee was undertaken as per University of Otago requirements for research proposals involving Māori. Ethics approval was received from the Central Health and Disability Ethics Committee in 2017 (17/CEN/94). The study protocol is available at: http://www.ANZCTR.org.au/ACTRN12617001469303.aspx, accessed 26 February 2021.

### 2.2. Data Sources

Data sources for this paper come from relevant policy documents and literature and interviews with staff and service users in adult acute mental health wards.

#### 2.2.1. Policy Documents

Key policy documents were sourced from the NZ Ministry of Health and included national-level specifications for the provision of mental health services in the acute mental health ward setting. To identify current philosophies of care in mental health with relevance to these settings in Aotearoa New Zealand, official ‘models of care’ were requested for the four case studies (described below) by an Official Informational Act request to the Ministry of Health. Of the four, only one DHB provided a model of care so this document was excluded and information with respect to philosophies of care were obtained from Ministry of Health publications.

#### 2.2.2. Case Selection

From the 20 publicly funded adult inpatient acute mental health wards, we selected four facilities to study, based on their age, representativeness and diversity [12]. After obtaining ethical approval, we approached those DHBs we were interested in including in our study. The first four DHBs we approached agreed to be case studies for our research.

#### 2.2.3. Interviews

The lead author, a social scientist and qualitative researcher, conducted all 85 interviews in the four adult inpatient mental health ward case studies between 2018 and 2019. Interview participant demographics are given in Table 1.

Interview questions were developed by the lead author (GJ) and are provided as Appendix A. The interviews were audio recorded, lasted between 30 and 90 min and were professionally transcribed. The majority of the interviews were conducted face to face on the ward with the remainder completed by phone. Among the questions, we asked them, from their perspective, what they thought the purpose of the ward was and how they would know if someone was ready to leave (these questions were designed to elicit perspectives on the purpose of the wards).

### 2.3. Analysis

We, GJ and JM and a research assistant, separately identified themes in the interview transcripts, then met together to discuss, refine and agree on the key common themes in the data following five step process outlined by Braun and Clarke [13].

## 3. Results

### 3.1. Defining Fitness for Purpose

Fitness for purpose has been defined as the fulfilment of a specification or stated outcome(s) and has been used to assess quality by establishing the extent to which the product/building or service fits this stated purpose [14]. Broadly, it offers two alternative priorities for specifying purpose, depending on the perspective. The first puts the onus on the provider for establishing purpose; the second locates it with the service user. These two models encompass top-down and bottom-up perspectives for establishing ‘purpose’.

The top-down mission-based fitness-for-purpose approach allows institutions to define their purpose through their mission and objectives and ‘quality’ is demonstrated by achieving these. Flexibility can be achieved for diverse facilities or diverse client populations as this definition allows for variability in institutions, rather than forcing them to be clones of one another [15]. With respect to acute mental health care in Aotearoa New Zealand, the mission can be established nationally, but then modified to suit the specific needs of the various local DHB client populations. This provides the flexibility for locally-based institutions, to address the needs of specific populations, while under the overall umbrella of the national provider.

The bottom-up service user definition of fitness for purpose defines quality as meeting the service user specifications, needs or requirements. In principle, in this model the service user is sovereign. The service user has requirements that become the basis of specifications for the service delivery and its setting, and the outcomes seek to reliably match these requirements. In this bottom-up model, success is defined by the performance of the service meeting service user objectives. To better understand the service and facility implications for each of these two approaches, each is put ‘under the microscope’ in the context of NZ.

### 3.2. The Mission-Based Approach to Fitness for Purpose in Acute Mental Health Facilities

The nature and scope of acute mental health care to be provided in Aotearoa New Zealand is broadly defined by the Ministry of Health in a series of national-level service specifications. These state that the purpose of care is to:

‘provide inpatient care for people in the acute stage of mental illness who are in need of a period of close observation and or intensive investigation, support and or intervention, where this is unable to be safely provided within a community setting, or a less acute inpatient service’ [16]

Additionally, many DHBs have their own models of care that describe the nature of services to be provided to their specific populations. Some of these refer to underpinning philosophies, cultures or principles of what mental health provision in their regions should ideally aspire to or look like.

In Aotearoa New Zealand, two key underpinning philosophies of mental health care exist, one is the Western ‘recovery model’ which follows the examples of the UK, Australia and North America and the second are the Indigenous Māori-based models, encapsulated in Mason Durie’s *Te Whare Tapa Whā* [17] and *Te Wheke* [18], which are used to define Māori health and wellbeing in an holistic manner (Figure 1 and Figure 2). Both the Western recovery model and the Te Whare Tapa Whā Māori model are enshrined in the Ministry of Health specifications for the delivery of acute mental health care through the three-tiered service specification framework.

A recent review of the implications of the Western recovery model for psychiatry concluded that ten factors were important for recovery [6] (Table 2).

Māori notions of ‘recovery ’and ‘well-being ’are located within a specific cultural, social and economic context where the extended family (whānau) and hapū (sub-tribe) is regarded as a source of strength, support, wisdom and identity [19]. See Figure 1 and Figure 2.

Thus, recovery for Māori needs to be seen within the context of the strength of a person’s identity within their whānau and hapu which recognises the importance of reconnection with the natural world and a person’s origins or whakapapa [20,21]. Recovery recognises that a person is nested within the generations, being mentored and guided by elders, and responsible for those that follow. Healing is supported by the love of children and the presence of elders. Recovery also recognises the spiritual realms within which Māori are located and acknowledges that particular ceremonies can mitigate the effects of spiritual harm.

### 3.3. Staff Perspectives on the Purpose of Acute Mental Health Services and Facilities

To obtain insight into how the mission-based, top-down, model of recovery is implemented, staff perspectives were obtained through interviews about the purpose of the mental health ward. Four main themes emerged from these interviews, which have been ordered in terms of their commonality. These were keeping people safe; containment; treatment and care; and, empowering the service user and offering therapies and encouraging reflection on life. In addition, to these most staff identified two functions which they did not consider part of an acute mental health facility; namely, recovering and detoxing.

#### 3.3.1. Keeping People Safe

Staff maintained that the primary purpose of the acute mental health facility was to keep people safe (not recovery). When staff talked about keeping people safe, they were referring to keeping the service users safe, from themselves, or from others, and sometimes keeping the community safe from the service user. Although some mental health service users are admitted to mental health facilities voluntarily, the majority are admitted under the Mental Health Act, 1992. The Act allows for people who are ‘mentally disordered’ be held under compulsion, in cases where they are deemed to be a serious danger to themselves or others; or to have seriously diminished capacity to care for themselves [s 2 Mental Health Act 1992]. In this context, ‘serious danger’ might comprise suicidal ideation or behaviours; the person’s behaviour may make them likely to be the victim of violence from others; or a particularly vulnerable person may be at high risk of being sexually exploited when affected by an abnormal state of mind. People are compulsorily admitted to an inpatient unit when they are judged by a psychiatrist to be mentally disordered requiring inpatient care and they are unwilling or lack the capacity to consent to admission. Thus, the legal mandate for admitting someone compulsorily unsurprisingly shaped perceptions of the purpose of the unit.

#### 3.3.2. Containment—Locked Wards

As part of keeping people safe, many, but not all staff, felt that acute mental health facilities needed to be locked facilities. The precise definition of a ‘locked’ ward curiously varied from ward to ward. In Aotearoa New Zealand, each acute mental health facility has at least two levels of care in terms of acuity—a higher needs or more acute area or ‘closed ward’ (with various names such as ‘high needs unit’, ‘high care area’, ‘intensive care unit’ or ‘retreat’) and, what was referred to as the ‘open ward’ (despite it often being locked), for those with lower levels of acuity. In three of the four cases studied, the ‘open ward’ was in fact locked—meaning that the door to the outside world could not be opened by the service user—it had to be opened by staff. In the fourth case, service users could leave the open areas of the facility and exit through the front door via a staffed reception area, but usually under their watchful eyes during normal working hours. However, many times when the facility was visited, staff had in fact locked the doors to the reception area, effectively making the facility ‘locked’ for at least some of the time.

#### 3.3.3. Treatment and Care

Once service users were safe and contained, staff identified the purpose of the adult acute mental health unit as a place to diagnose, observe, treat and care for people who were acutely mentally unwell or experiencing a mental health crisis, which could not be managed in a less restrictive setting. Staff reiterated that the mental health unit provides an *acute* service, with a few staff likening the acute mental health ward to a mental health emergency department. Mental health wards were viewed as short-term mental health crisis intervention centres.

In cases where staff knew the service user, their aim was to stabilise the person’s mental state and get them back to their ‘baseline’. Observation, monitoring, and the prescribing and adjusting of medications were tools used to achieve this aim. In cases of new first-time admissions, the staffs’ aim was to observe, diagnose, treat, monitor, and formulate a treatment plan based on available treatment options. Other staff talked of ‘getting’ service users ‘balanced’, getting them ‘through their initial crisis’, or getting them ‘over their acute phase’, by minimising symptoms and managing risk factors.

The importance of sleep, rest and relaxation to facilitate the recovery of mentally unwell service users was emphasised by many staff. While staff recognised that in reality the acute mental health ward could often be chaotic, unpredictable and noisy, they noted that it *should* be a quiet restful place offering respite from stresses, and a place that encourages rest, relaxation, and sleep. It should also, according to some staff, provide nourishment, food and love. Staff aspirations for the ward environment were that it should be an attractive, healing, comfortable place, that facilitated the involvement of family and whānau and reflecting the needs of the service user, cultural and otherwise.

#### 3.3.4. Empower Service User, Offer Therapies and Encourage Reflection on Life

Some staff felt that the ward environment *should* be one that empowered service users to speak for themselves and should get them to a place where they (the service users) could engage and reflect on where to next. Many felt that the ward environment should facilitate self-responsibility and help people to maintain simple routines, such as making beds and preparing food as well as obtaining insights into their illness and accepting their treatment.

Staff commented that the ward should offer therapy and provide support and tools for coping, although they lamented that few therapies were in fact available on the ward. In all four wards studied, treatment options centred on prescription medication, some basic occupational therapy activities, such as art therapy, with two wards offering the occasional music or pet therapy, and very little in the way of talking therapies apart from talking to the nurses or occasionally the social worker. Most wards relied on a resident psychiatrist, psychiatry registrars in training, one or more occupational therapists, social workers, nurses and care assistants. This therapeutic model is perhaps best summarised as a psychiatric dominated medical model. No wards had a psychologist. Staff regretted this lack, suggesting that a number of the service users would benefit from psychological therapies especially cognitive behavioural therapy (CBT) and dialectical behavioural therapy (DBT).

Some staff felt that the function of the ward was to support recovery, to provide long-term strategies, to work with service users through their recovery journey and to provide psychological resources such as talking to them as part of treatment. However, a small number of staff, predominantly nurses, suggested it was not their job to provide counselling.

In defining the functions of an acute mental health facility, some staff clarified those functions that did not fit with their notions of mental health ward functionality; namely recovery and detoxification.

‘Recovery is not the goal of the acute mental health ward’

For many staff, recovery was *not* considered a function of the acute mental health care facility. Many staff deemed people to be too unwell to engage in therapies on the ward and recovery was considered to be something that would happen back in the ‘community’. They felt the function of the acute mental health facility was to facilitate the development of an individual plan for each service user to be implemented in the community, re-connecting them with the community and their whānau and family; linking them with their community mental health team, and involving them with other community resources. So, having a plan for recovery, rather than recovery itself, was viewed as the goal.

‘Not a detox unit’

Some staff considered that detoxification was not a function of an acute metal health ward, despite the fact that they were providing a place for people experiencing abnormal mental states due to intoxication or withdrawal from illicit drugs such as methamphetamine and synthetic cannabis. These staff felt that there should be separate facilities for detoxification—and that service users with substance use disorders were behaviourally different to those with mental illnesses. Many staff believed that people presenting with drug-induced psychosis were often more aggressive and irrational and needed different treatment to those with other psychotic disorders. Staff lamented (wrongly), however, that there were no units for substance-induced mental health disorders, so such people under the influence of illicit drugs often ended up in acute mental health wards.

### 3.4. Measurement of Performance: Model of Care in a Mission-Based Model

To better understand the implementation of the recovery model in a mission-based approach, it is useful to interrogate the means for assessing the quality of outcomes. In the mission-based approach these are embodied in key performance indicators (KPIs) combined with reports on health outcomes for the ward, which are garnered from clinical data. The DHB sector has clinicians in acute mental health care settings who assess the social and health functioning of service users with severe mental health problems in their care, using the 12-item Health of the Nation Outcome Scale (HoNOS) [22] The scale is used on two occasions, on admission and on discharge to measure change [23]. The scale includes items such as aggression, non-accidental self-injury, problem drinking or drug taking, problems with depressed mood, relationships, daily living, living conditions and work, rating each domain on a 5-point item of severity. Some DHBs supplement this with other feedback from service users via service user forums, as well as complaints processed, although these are under developed [3].

As the Ministry of Health mission identifies service user perspectives as forming an important part of their recovery philosophy, performance measures must extend beyond the standard KPIs noted above. For the service user, measures of performance are enshrined in quantitative patient satisfaction surveys which the Ministry of Health has commissioned for some time. More recently the Health Quality and Safety Commission (HSQC) has also begun to gauge staff views on quality of service provision in the acute mental health setting by survey [24]. These surveys rely on Likert scales with little qualitative information. The HQSC has surveyed patient experience, examining four domains: communication, partnership, coordination, and physical and emotional needs. Although it has a very low 3% response rate, results find that people were more positive about community based mental health services than in-patient mental health services. https://www.hqsc.govt.nz/our-programmes/health-quality-evaluation/publications-and-resources/publication/3936/. accessed 26 February 2021.

One challenge with interpreting the results of these surveys is that they do not always differentiate between acute mental health in-patient service users and community mental health service users (and they should). Some work has also been performed around the development of Māori-centred mental health outcomes although this does not appear to have been incorporated into mental health service user research or surveys [25]. A final method of quality assessment is undertaken by the Ombudsman’s office using unscheduled site visits where staff and patients are interviewed, and facilities observed.

### 3.5. Architectural Implications and Design Criteria for a Mission-Based Model

In Aotearoa New Zealand, Ministry of Health service specifications have been developed in a tiered system. The objectives of Tier 1 revolve around adaptability to meet the unique needs of specific population groups and individuals [16]. This is to be achieved through ‘being flexible around service delivery settings in both urban and rural areas and adaptable to service users individual circumstances and needs, including cultural and spiritual needs’ [16]. Tier 1 also establishes that the ‘overarching aim of the health and disability sector is the improvement of health outcomes and reduction of health inequalities for Māori’ actively involving tangata whenua (Indigenous Māori) in planning for services. Tier 2 and Tier 3 separate adult acute inpatient services from other acute services and care packages by age, ethnicity and service type. These tiered objectives are then distilled into criteria for the design and refurbishment of facilities rather than using the recovery models.

The criteria for the design and refurbishment of psychiatric acute inpatient care facilities are laid out in the 2002 Ministry of Health’s *Criteria for the design and refurbishment of psychiatric acute and intensive care facilities* [10]. The document advises of the Ministry’s expectation that the DHBs will provide safe and effective facilities that are consistent with the National Mental Health Sector standard (NZS 8134) (https://shop.standards.govt.nz/catalog/8143%3A2001%28NZS%29/view) accessed 26 February 2021.

The criteria are clear that while there is no expectation that facilities comply with a specific design brief, there are a number of principles and general requirements that must be met. However, while much of the content of this statement document is claimed to be based on lessons learned from actual examples of inadequate design and planning [10], there is no literature available that shares these ‘lessons learned’ [10]. The document acknowledges that ‘facility design may act as a tool or impediment to recovery’ [10] and that ‘well-designed facilities are a necessary element in delivering effective care’ [10]. It also proports that ‘poorly designed facilities can make it more difficult to provide effective care, recruit and retain skilled staff and deliver services with a recovery focus’ [10]. As guidance, the document includes the following table to set out what the facility design needs to accommodate (Table 3).

Finally, the document contains a section ‘Design specifications’ which provides more detailed advice for the achievement of three specific goals, namely (1) to design to promote autonomy and choice; (2) to design to provide privacy; and (3) to design to support safety [10]. In this section, more specific direction is given with respect to the achievement of these specifications. For example, the need for ‘adequate, well laid out, outdoor space for patient enjoyment and exercise’ [10]; ‘the provision of single bedrooms either with ensuites or in close proximity to gender specific bathroom facilities’ [10]; and ‘ability to maintain a high level of observation, both sight and sound, and free access to all areas’ [10].

### 3.6. Measurement of Performance

To ensure that facility designs are adequate and appropriate (fit for purpose), the Ministry requires ‘evidence of research and assurance that the quality of the materials used, and the designed spaces are adequate for the care of sometimes very disturbed patients’. However, the document notes specifically that ‘the key to success is realism about available budget’ [10]. While budgets are mentioned in several places, these ‘should not override the basic design philosophy, which is aimed at assisting staff to provide quality psychiatric acute and intensive care services [10]. It is unclear how the recovery model is addressed in these specifications.

Following construction, fitness for purpose is reportedly assured through quality assessment procedures. In theory, these are completed by the institution demonstrating they fit either externally-prescribed standards (such as those specified by the regulatory or professional bodies) or its own objectives, as specified, for example, in its values and mission statement. An institution that embraces a mission-based fitness-for-purpose approach to standards will typically adopt objective criterion-referenced assessment criteria for the building, rather than subjective norm-referenced assessments. This is seemingly to address different interpretations. For example, homeliness may mean different things to different people, and for different locations, cultural orientations and populations. Sleeping and eating arrangements might look very different depending on cultural orientation. The challenge of course is in the establishment of the criteria.

### 3.7. The Service User Approach to Fitness for Purpose

To obtain a service user perspective on fitness for purpose, mental health service users from the four case study examples were interviewed. Coding from the interviews with mental health service users on the ward found that perspectives centered around five main themes, which have been ordered in terms of their commonality. These were: providing respite from their symptoms and stress; providing safety and security without being locked up; providing them with both company and privacy while recovering; meaningful activity; and the development of a mental health after-care plan for re-joining their community. With the exception of a handful of service users who felt the ward was a good place to be as it kept them safe from harming themselves, the vast majority expressed the view that being on an acute mental health ward was not a place they wanted to be.

When service users were asked what they wanted to achieve from being on the ward, they overwhelmingly said they wanted to get out and go home (or find a home in cases of homelessness). To be discharged, they knew that they had to ‘get well’ or at least get better than they were on admission. ‘Getting well’ meant different things to different service users. For some, it meant ‘stopping voices’ or ‘bad thoughts’, others talked about ‘sorting out their life’, ‘getting life back on track’, getting back to their ‘normal’. Some service users talked about wanting to ‘fix their brain’ or ‘find a cure’, while others talked about wanting to reduce their symptoms, decrease suicidal thoughts, manage anxiety or depression and find peace of mind.

#### 3.7.1. Respite from Their Symptoms and Stress—Wanting Sleep and Medication

The main function of the acute mental health ward, from the perspectives of services users, was to provide much needed respite from mental distress. Sleep and medication were both key to this respite. Many talked of having suffered from a lack of sleep prior to the crisis that precipitated admission. Some attributed the lack of sleep to their mental illness, while others felt that the lack of sleep exacerbated their illness. Service users felt that the function of the ward was to provide a peaceful restful environment conducive to good sleep. Sleep offered them important respite and a number of service users mentioned wanting medication to help them sleep and to help them manage their anxiety. Service users felt that the function of the facility was to provide them with desired stability in their medication, but also relief from the adverse effects of their medications.

#### 3.7.2. Safety and Security but Not Locked Up

Like staff, service users placed a high priority on safety. Most wanted to feel safe and secure. A number were able to recall times when they did not feel safe in the ward. Some were frightened by the actions of other service users on the ward, especially those who were behaving loudly or aggressively.

Service users had a range of views on locked wards. There were a group of service users who were vehemently opposed to the doors of the ward being locked. Many people felt ‘caged in’, or like they were in prison. However, a minority of service users expressed an acceptance of the locked ward as in their own best interests, usually referring to feeling safer from their own risk of self-harming.

#### 3.7.3. Company and Privacy

Service users generally wanted a mix of company and privacy. Most wanted someone to talk to who could advise them on ‘how to get better’ or when they were ‘making bad choices’. At the same time, most also wanted privacy, from staff or other service users, or both. Privacy often meant acoustic privacy in their rooms away from staff, and also visual privacy from staff and service users peering in or walking into their bedroom or the bathroom. A number complained that they could not lock the door to their room when they were inside to keep others out, and could not lock the door to their room to keep their things safe, once they left their room to go elsewhere in the ward.

#### 3.7.4. Meaningful Activity

Service users complained of a lack of activities on the ward and a very limited range of therapies on offer, apart from medication. They considered that an important function of an acute mental health ward was to provide them with meaningful activities that contributed to their wellbeing and successfully building and implementing their recovery plan. A common experience was that such activities were notably absent.

#### 3.7.5. Developing a Plan

Many service users talked about wanting a plan for the future. For some, this was as basic as being able to *see* a future; for others, it meant envisioning an ‘okay’ future. Most service users wanted an explicit plan for life back in the community. They expected that such a plan was part and parcel of the purpose of the service being provided in the acute mental health ward and that they would be released with a plan in place to connect them with a case worker and relevant health professionals to provide support in the community. The plan would also help them with accommodation issues, maintaining health appointments, follow-up on recovery and life plans generally, as well as connecting them with someone to whom they could talk. Some wanted assistance with finding or returning to work, saving money, having things to do, and both creating and achieving goals.


**Implications of service user narratives**


Based on service user narratives, three key stages of occupancy need to be accommodated in both the delivery of care as well as the design of the facility. Based on interview themes, the following three stages were developed:

Stage one (entering)

Preventing harm, andRemoving people from the community.

Stage two (stabilising)

Diagnosing, stabilising and treating,Providing respite from symptoms of mental distress, reducing the risk of harm, andFacilitating ease of care and establishing the therapeutic relationship.

Stage three (exiting)

Once stable, preparing for re-entry into community,Contributing to wellbeing, dignity, respect (both staff and service user), andReturning people to the community.

These are similar to the Australian synthesis of recovery narratives (see Table 4) that identifies five phases of recovery, of which three are related to acute mental health and two to community mental health [26]. Andresen et al. [26] point out that the type of help and support that promotes recovery will differ depending on the stage of recovery. Promoting self-management for someone in the Moratorium stage, for example, may give rise to feelings of abandonment.

### 3.8. Measurement of Performance: Model of Ccare in a Service User-Based Model

Measures of performance in a service user model are somewhat more complex than the mission-based model as they must be independently obtained. We were unable to find many examples of qualitative inquiry undertaken by official advocacy groups and there appeared to be little quality assessment driven by service users to assess what they think is important in the provision of acute mental health care and its setting. Two exceptions to this include a study of Māori recovery narratives [27] commissioned by the Mental Health Commission and the second, a crowdfunded personal story-based publication ‘The People’s Mental Health Report’ which did not focus directly on provision of care in the acute mental health ward [28].

### 3.9. Architectural Implications and Design Criteria for a Service User-Based Model

Through an analysis of recovery themes from both qualitative narratives from the literature [7] as well as those from our interviews, the requirements for physical environments were distilled into five key themes. These themes align with the overarching CHIME model, namely connectedness, hope and optimism, identity, meaning and purpose and empowerment, with one exception, safety and security, which we have added to make CHIMES. The themes also align with the themes from our interview data and the two Māori models (shown in italics in Table 5). It is important to note that while the themes may align, cultural differences create different service implications and different architectural implications.

The theme of connectedness extends to several levels of connection, the fostering of relationship development and support from others, the facilitation of peer support and support groups, and the connection to the wider community. For heightened quality of life while in care, service users described the need for the friendship of other service users, to be able to host visitors, and to productively live with others. To facilitate these relationships, facilities need to provide spaces for friends, family and whānau to visit, to provide common rooms where productive activities can take place such as doing laundry, preparing food, talking privately on the phone or via the internet and participating in community building activities, such as online courses and discussion groups.

Creating an environment which fosters hope and optimism is deemed to be essential to aid in the motivation for change and the belief in the possibility of recovery. Recovery entails having dreams, aspirations and positive thoughts for the future. Access to nature and natural light have been widely reported as beneficial [29] Warmth and sunshine in a facility balanced and harmonised with both calming and stimulating spaces can help to reduce distress and relieve pain. Attention to all aspects of the qualities of the indoor environment have shown to produce beneficial outcomes [30].

The establishment of identity and the redefinition of a positive sense of identity together with the journey of rebuilding it are part of the recovery process. Design can aid in allowing the personalisation of private spaces, storage of personal items and the maintenance of possessions and nostalgic objects. Identity can also be fostered by providing options giving autonomy and choice. To address the unique needs of individuals, dualities of diversion and contemplation demand dynamic interactive spaces that allow for escape and stimulation as well as the opportunity to sit quietly and alone, meditate, gaze on nature, suggest open-ended imagery which is rich for the senses. Social interaction can be fostered through the design of a multi-use space that has a broad-based appeal but can be balanced with nurturing spaces that are family centred, with culturally appropriate imagery.

Fostering meaning and purpose can be aided with facilities, such as small meeting rooms to allow the shared exploration of the meaning of the mental illness experience with other service users, with rooms designed for contemplation, confession and spirituality and prayer as well as spaces for staying fit and able, contributing to the community or a special interest group. The enabling of meaningful activities, continuing on with the familiar but also trying out the new activities.

The provision of suitable architecture can empower the individual, allowing them to focus on their strengths, facilitate personal responsibility and control over life by allowing service users to maintain and personalise their space. People are often confused and afraid when they enter a mental health ward, not sure where they are allowed to go. Having a design that makes sense, is well laid out and signposted helps allay that fear and disorientation. With well-designed facilities, greater control over spatial use is also possible. Through the inclusion of spaces for making choices, such as laundering, shopping/selecting, cooking, moving around, showering. Details are important.

Design for a choice of activities to occupy time: learning, writing, making art and music, gardening, cooking, animal husbandry, performance, religion, hobbies, reading, and strategic sports. A further benefit of designing opportunities for engagement in interesting ideas and activities is that such interests are infectious (so there is a greater social benefit) and they are ongoing—vulnerable people can take these ideas and interests with them wherever they go, and these skills will prove protective against unwanted automaticity, paranoia, and other symptoms. This is the true meaning of recovery-centric design.

Finally, fostering safety and security can extend beyond the direct risks of self-harm, violence and vandalism and ligature removal to a focus on providing good visibility, with wayfinding, access and egress designed so that individuals do not feel trapped [30,31,32,33,34].

### 3.10. Measurement of Performance

There were no apparent measures of facility performance undertaken by the service users. The Ombudsman’s reports include aspects of facility performance, but are largely confined to the absence or evidence of types of space, such as courtyards and family rooms. The Ombudsman inspectors also informally interview service users and staff in their impromptu visits. The Office of the Auditor General has conducted some ad-hoc reports, for example a recent report investigated what happened after discharge from acute mental health care. More directly, the first national survey, Ngā Poutama Consumer, Family and Whānau Experience Survey was undertaken in 2019 with 267 people who had recently used DHB inpatient or community mental health and addiction services [24]. While the response rate was disappointing (3.3% of service users), questions included whether families and consumers were treated with respect, whether cultural needs were respected and whether consumer values and beliefs were incorporated into care and support plans, opportunities for reporting on facilities were included with a focus on quality and safety and culture. More positive responses were evident for those recently using community based services than those using inpatient services and, those under compulsory treatment were less likely than people under voluntary treatment to agree that they felt warmly welcomed into the service; they also felt they were less able to have a support person with them during sessions with staff.

## 4. Discussion

This research investigated the question of ‘fit for purpose’ to better understand the performance requirements for acute mental health facilities in terms of the models of care and the facilities themselves. It found that more than one definition of ‘purpose’ exists and that even within one definition, there was no widespread agreement on what the purpose of the acute mental health facility was. For example, some were of the opinion that serving all service users experiencing mental health problems in acute care was part of the purpose, while others maintained the purpose was not to function as a detoxification unit and that service users with substance-induced disorders had different requirements than those with primary mental illnesses and should have bespoke facilities. Similarly, some maintained that an acute care unit was an element of a hospital while others felt it was more of a halfway space between hospital and the community.

Two approaches to fitness for purpose were then examined, the mission-based top-down model which derives from organisational goals and objectives and a bottom-up service user model which derives from consumer satisfaction. While the two approaches share a desire for recovery, there are significant differences in the strategies undertaken for quality assurance of both the models of care and the facilities that support levels of care.

Taking the mission-based approach, the purpose of the facility is for recovery. While institutionally, the mission does not differentiate between causes of illness, there are clearly issues about entitlement among staff and service users differentiating between ‘deserving and undeserving illnesses’. In the mission-based model of care, the literature finds that recovery is determined by ‘shortened lengths of stay, decreased rates of medication refusal, reduced verbal and physical aggression, reduction in depression and self-harm, reduction in medication use, and improvements in mood, social interaction, staff and service user satisfaction and sense of safety’ [35]. Inpatients are assessed for recovery through regular mental health examination and discussion with family, whānau and service users.

The service users’ model of recovery provides more options than simply length of stay, conditions of medication and events of self-harm. Measures are more qualitative than quantitative as they seek to gauge: service users’ levels of hope and optimism; their connectedness to peers, significant others, and the community; their identification as positively contributing individuals as opposed to being defined as a ‘psychiatric patient’; and their levels of empowerment. Meaning and purpose involve the engagement with meaningful activities, a rebuilding of life and social goals, a sense of spirituality and an understanding of quality of life. Finally, empowerment involves choice and the ability to determine one’s own good outcomes and reclaim control over life, being involved in decision making and taking personal responsibility. These types of outcomes can only come in stages, as a person progresses through their journey to recovery. While the process is not linear, three stages can be established; the entry, the stabilisation and the exit.

Within the service user’s model of recovery, there needs to be an assessment of recovery that addresses Māori cultural differences, where recovery is established by whānau, rather than by clinicians. While themes for recovery align across cultures, the implications for both service delivery and architecture are not compatible and the extent to which Māori models of health and wellbeing have been successfully integrated into treatment programs appears limited [36]; however, while there is some consistency between Māori health models and the CHIME model for recovery, there is less alignment with respect to actual treatment. For example, the Western-based cognitive behavioural therapies where one seeks to change dysfunctional thoughts and beliefs thorough inward thinking, contrasts with the whānau concept set out in Te Whare Tapa Whā, which describes overly self-referential thinking as unhealthy [17]. Mental health is not just about thought patterns, there are environmental, existential and spiritual factors which can be respected by acknowledging Māori medicines, such as rongoa (traditional Māori medicine); massage-based treatments, such as mirimiri; and karakia (prayer). Similarly, cultural ceremonies can help mediate the concerns of spiritual harm. Finally, the purpose is for service users to reintegrate with the community; however, the duration of admissions vary and, according to staff interviewed, frequent returns are common for some. There did not appear to be any evaluation of both the care model and the facility in terms of examining the contributions to returns.

This multidisciplinary mixed methods qualitatively driven research involved interviews with staff and service users of four different inpatient wards. All facilities were visited and studied in depth. The limitations include the fact that the units selected were only a subset of the 20 adult mental health wards in Aotearoa New Zealand. Findings may not be generalisable to other mental health units or to units outside Aotearoa New Zealand. The measures of performance, however, tell us what is important to the institutions responsible for mental health as what is valued is measured.

A number of important findings were derived from this research. The first finding was that the purpose of an acute mental health facility was neither clear nor universally accepted. There were a range of perspectives on what the purpose should be and whose interests it should serve. While current definitions that pair mental health with addiction is clear in the Ministry of Health guidelines, this has not been widely accepted among service providers or service users. There is work to be performed either engaging and uniting these groups or alternatively, modifying the guidelines to address the difference between substance use disorders and other forms of mental health illness.

Secondly, our research found that the meaning of recovery was similarly not universally agreed upon. There were different perspectives on recovery held between providers, service users, staff and cultural advisors. Further research is required to obtain alignment between Māori and non-Māori models of wellbeing and how specifically this might translate into both models of care as well as facility design. In addition, we find that acknowledgement of stages of recovery which all have different objectives and which require different measurements are key to facilitating recovery. Three distinct stages of recovery were identified, each of which had relatively unique functions. First, there were the immediate functions required of entry into the facility, then there were the functions relating to taking back control, and finally, those functions relating to exiting the facility. This would indicate different facility requirements as well as different models of care for each stage.

The third finding of our research was of an international framework/model, CHIME, that we believe could accommodate both Māori and non-Māori recovery objectives and allow for the translation of these into both models of care and facility design directives. Our research finds that adopting a recovery model enshrines both mission-based and service user approaches to providing quality care.

Our fourth finding was that some additional requirements surfaced but these differed between staff and service users. Both groups expressed keen concerns for personal safety; however, their views differed on how to achieve this. Generally, staff endorsed locked wards, but most service users sought other means of safety and security without locked doors. Staff needed ease of diagnosis, observation, treatment and care, while service users sought respite from their symptoms and stress, wanting sleep and medication. Both staff and service users sought meaningful activities, for service users this was to alleviate boredom, but for staff it was to provide therapies and encourage reflection on life. Finally, service users sought a balance of social connection or company with the need for privacy.

The final contribution of this paper has to do with how recovery is measured. Current mental health policies highlight the agenda for those who run our mental health establishments and are mission based rather than addressing the recovery experience and trajectory of the service users. All measurements are currently quantitative (Likert scales are an attempt at qualitative methods) and fail to effectively capture the achievement of quality of both care and facility.

There is extensive research to support the importance of the role of architecture to mental health; however, much of it has been distilled to create ‘optimum healing environments’ rather than specifically focusing on the goals and objectives—the purpose, of the facility.

## 5. Conclusions

This research investigated the ‘fitness for purpose’ of acute mental health facilities in Aotearoa New Zealand. The definitions and perspectives on the purpose of an acute mental health facility have significant bearing on the strategies for models of care and their implementation as well as the architectural design. It found an important lack of clarity and agreement in terms of the purpose of the acute mental health facility and the definition of recovery. Further work is required to align staff and service users with Ministry of Health understandings of the purpose of the acute mental health facility and the means for achieving recovery goals. While to be discussed and contested, we find that both the mission-based recovery model, the service user recovery model and the Māori service user health models align with general definitions of fitness for purpose. However, in New Zealand, purpose is focused on wider goals of service user recovery. Unfortunately, considerations of facility design and measurement of outcomes are underdeveloped in extant models of care.

Finally, a transdisciplinary approach to research is essential for bridging the gaps between overarching objectives and a holistic implementation that encapsulates models of care and facility design which can address the concerns of service provider and service user. The voices of these key groups are essential and can form meaningful direction through co-design.

## Figures and Tables

**Figure 1 ijerph-18-02343-f001:**
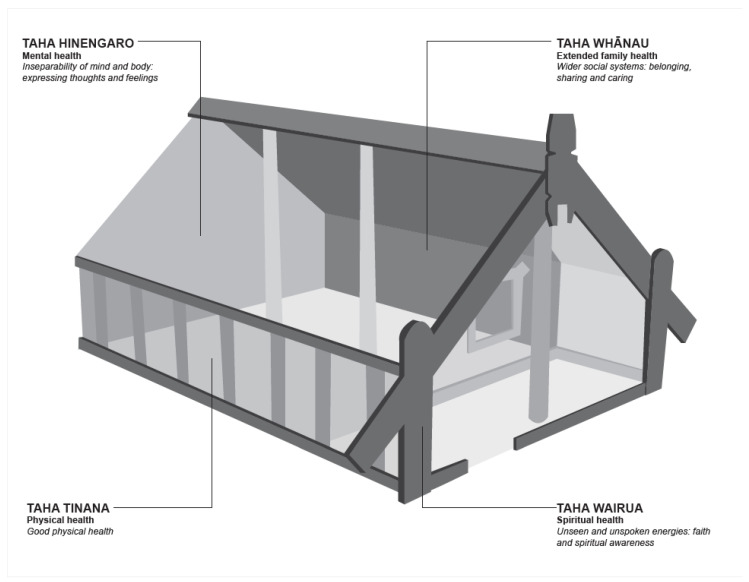
Te Whare Tapa Whā. Source: Redrawn from Durie M. 1994. Whaiora, Maori Health Development. Auckland: Oxford University Press.

**Figure 2 ijerph-18-02343-f002:**
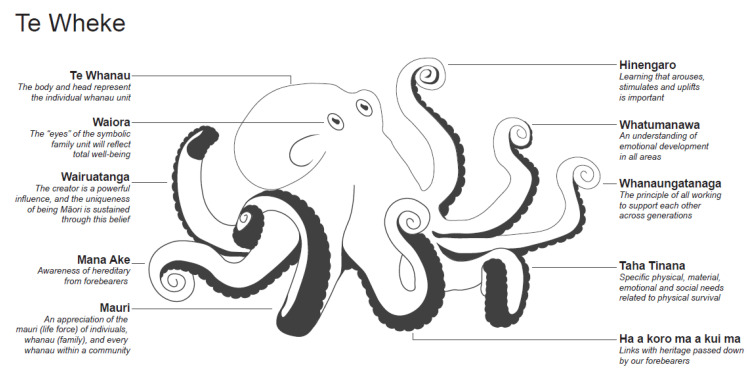
Te Wheke. Source: Redrawn from Pere RT. 1997. Te Wheke: A Celebration of Infinite Wisdom. NZ: Ao Ako Global Learning.

**Table 1 ijerph-18-02343-t001:** Interview participant characteristics.

Participants	Ward A	Ward B	Ward C	Ward D	Total
Service users	10	11	12	10	43
% female	50%	45.5%	58.3%	50%	51.2%
Staff	9	13	11	9	42
% female	44.4%	53.8%	63.6%	100%	64.3%
Total	19	24	23	19	85

Of the 43 service users, 34.9% were Indigenous Māori (and the highest proportion of Māori were in Ward D) and 16.7% of staff (*n* = 7) were Māori. Staff included in the interviews were from a range of professions, including nurses, nurse aids, occupational therapists, psychiatrists and clinical team leaders.

**Table 2 ijerph-18-02343-t002:** Consensus statement for recovery [6].

Principle	Description
Self-direction	Consumers lead, control, exercise choice over, and determine their own path of recovery
Individualised and Person Centred	There are multiple pathways to recovery based on the individual person’s unique needs, preferences, and experiences
Empowerment	Consumers have the authority to exercise choices and make decisions that impact their lives and are educated and supported in so doing
Holistic	Recovery encompasses the varied aspects of an individual’s life including mind, body, spirit, and community
Non-Linear	Recovery is not a step-by-step process but one based on continual growth with occasional setbacks
Strengths Based	Recovery focuses on valuing and building on the multiple strengths, resiliency, coping abilities, inherent worth, and capabilities of the individual
Peer Support	The invaluable role of mutual support in which consumers encourage one another in recovery is recognised and promoted
Respect	Community, system, and societal acceptance and appreciation of consumers—including the protection of consumer rights and the elimination of discrimination and stigma—are crucial in achieving recovery
Responsibility	Consumers have personal responsibility for their own self-care and journeys of recovery
Hope	Recovery provides the essential and motivating message that people can and do overcome the barriers and obstacles that confront them

**Table 3 ijerph-18-02343-t003:** The list of needs that the facility design must accommodate [10].

Privacy for Patients	Support for Clinical Functions and Observation
Safety and security for patients	Safety and security for staff
Domestic scale (homeliness)	Welcoming public spaces
Domestic feel	Durable materials
Least restrictive environment	Secure space for detainment
Gender safe, age safe, culture safe places	Space for inclusive social interactions
Retreat, sanctuary	Choice of spaces for different activities
Autonomy	Sense of belonging
Specifically designed intensive care facilities	Flexible spaces

**Table 4 ijerph-18-02343-t004:** Phases of recovery identified from recovery narratives [26].

Setting	Stage	Characteristics
Acute mental health	Moratorium	characterised by denial, confusion, hopelessness, identity confusion and self-protective withdrawal.
	Awareness	the first glimmer of hope for a better life, and that recovery is possible. This can emerge from within or be triggered by a significant other, a role model or a clinician. it involves a developing awareness of a possible self, other than that of mental patient.
	Preparation	the person resolves to start working on recovery, e.g., by taking stock of personal resources, values and limitations, by learning about mental illness and available services, becoming involved in groups, and connecting with others who are in recovery.
Community mental health	Rebuilding	the hard work stage, involving forging a more positive identity, setting and striving towards personally valued goals, reassessing old values, taking responsibility for managing illness and for control of life, and showing tenacity by taking risks and suffering setbacks.
	Growth	[May also be considered the outcome of the previous recovery processes] whether or not symptom free, the person knows how to manage their illness and stay well. Associated characteristics are resilience, self-confidence and optimism about the future. The sense of self is positive, and there is a belief that the experience has made them a better person.

**Table 5 ijerph-18-02343-t005:** The list of needs that the facility design must accommodate [7], Bird et al. (2014) modified by the authors for architecture.

Theme	Service Implications	Architectural Implications
Connectedness	Peer support and peer groups Relationships Support from others Being part of the community *Company and privacy*	Spaces for family and friends to visit Common rooms where productive activities can take place, such as, laundry, preparing food, repairing clothing Participating in online communities Undertaking community building activities Single user rooms and option for whānau/family sleeping Design of waiting rooms for privacy Waist-high partitions in communal spaces Acoustical treatments Elimination of long corridors Sociopetal furnishing layout
Hope and optimism	Belief in possibility of recovery Positive thinking and valuing success Having dreams and aspirations Hope inspiring relationships Motivation to change *Respite from symptoms* *Developing a plan*	Access to nature Access to natural light Mix of calming and stimulating environments Full spectrum artificial lighting Quality/supportive environments for staff to decompress Warmth and homeliness Indoor air quality Easy to clean surfaces Culturally appropriate reception area
Identity	Rebuilding or redefining a positive sense of identity Overcoming stigma Dimensions of identity *Developing a plan*	Options giving autonomy and choice Stimulation/solitude Family-oriented/individual-oriented social interaction Maintenance of possessions Storage of nostalgia items Cultural appropriate courtroom * design Culturally appropriate food preparation and dining areas
Meaning and purpose	Meaning of mental health experiences Spirituality Meaningful life and social goals Quality of life Rebuilding of life *Company and privacy* *Meaningful activity*	Rooms for contemplation, spiritual connection and prayer Family support areas Space to stay fit and able Space for privacy Space to contribute to the community Space to continue meaningful activities Opportunities to try new activities
Empowerment	Personal recovery Control over life (transparent rules on the ward) Focusing on strengths *Meaningful activity*	Selection of options to focus on strengths Shopping Making art and music Learning Hobbies Gardening, cooking, caring for animals Moving around Showering, personal care Greater control over position of the bed, lighting (including dimmers), sound (music and television), natural light, air temperature Orientation and wayfinding
*Safety and security*	*Safety and security* Patient interactions Staff behaviours and attitudes Non-consensual treatment	Unobtrusive technology Glass partitions Direct sight lines Clear spatial delineation Attention to spatial ordering Secure private room People rather than keys/relational security Reduction in locked doors Provision for security of personal belongings

* Acute mental health wards in Aotearoa New Zealand have their own courtroom.

## Data Availability

The data from the interviews are not publicly available.

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
