# Peer review of "Fit for What Purpose? Exploring Bicultural Frameworks for the Architectural Design of Acute Mental Health Facilities"

_ijerph, 2021, doi:10.3390/ijerph18052343_

Round 1

Reviewer 1 Report

The article ‘Fit for What Purpose?  Developing a conceptual framework for the
design of acute mental health facilities’ by Jenkin and co-workers investigated the ‘fitness for purpose’ of acute mental health facilities in NZ. Jenkin and co-workers found that the definitions and perspectives on the purpose of an acute mental health facility have significant outcomes on the strategies for primary care model, their implementation and architectural design. They found an important lack of clarity and agreement in terms of both the purpose of the acute mental health facility as well as the definition of recovery. Their research investigated the ‘fitness for purpose’ of acute mental health facilities in NZ. They also found that mission-based recovery model, the service-user recovery model and the Māori service user health models align with general definitions of fitness for purpose. This work falls within the radius of the aims and scope of the journal, ‘International Journal of Environmental Research and Public Health’, as it highlights the need of service user recovery as one of the prime considerations for models of care, facility design and measurement of outcomes. This kind of service user recovery model is underdeveloped in many countries and will be beneficial in the long run in the public health care system. Finally, they also emphasized on the need of a transdisciplinary approach to research for bridging gaps between overarching objectives and a holistic implementation that underlies models of care and facility design and have strong potential to address the concerns of service provider and service user. This will in turn beneficial for the growth and efficient functioning of public health care system. The article is well written and the use of English language is understandable. The study has been well designed and meets good technical standards. The methods, tools, software and reagents are well described and will aid in producing reproducible data by independent groups.

However, there are few minor points that need to be addressed:

  1. Spelling of Discussion needs to be changed. The current version mentions it as Discution.
  2. The results should have some quantitative representation. Currently the results are all in table form and very descriptive.
  3. The study subjects are low – 20. A higher number representation is better along with a nice quantitative scale.

Author Response

However, there are few minor points that need to be addressed:

1. Spelling of Discussion needs to be changed. The current version mentions it as Discution.

We could not find the spelling mistake noted by this reviewer. We used both the Find function as well as checking our document and could not locate this.

2. The results should have some quantitative representation. Currently the results are all in table form and very descriptive.

We have added a table into the Methods section under the subheading Interviews (P3, Line 125), to address the concern regarding a lack of quantitative representation.

3. The study subjects are low – 20. A higher number representation is better along with a nice quantitative scale.

We conducted 85 interviews.  The number 20 refers to the number of facilities in New Zealand (the case study sample frame) from which we selected 4.  We conducted 85 interviews from individuals at these 4 institutions.

Reviewer 2 Report

Thank you for the opportunity to review this manuscript. The authors investigated the ‘fitness for purpose’ of acute mental health facilities in NZ with qualitative research methods. According to these findings, they suggested the importance of bridging the gaps between overarching objectives and holistic care and facility design which can address the concerns of the service provider and service user. This research filed has been rarely examined in the academic field. The novelty of this study should be highly appreciated. Although this paper contains a wealth of information, the authors should strive to facilitate the reader's understanding.

I showed several points of requests as follows.

- The Introduction carefully describes the people, organizations, statements and guidelines involved in acute mental health care. The authors use diagrams to help experts in other countries better understand the relationship.

- In the Method section, the current description does not allow other researchers to replicate this study. Did the lead author conduct an analysis of 42 interviews with staff in 4 acute mental health wards and 43 interviews with service users all by only one researcher? The analysis should be performed by at least 3 researchers to ensure objectivity and validity. Did you use the question list or interview guide in the interview? The list or guideline should be submitted as supplementary information.

- In the Results section, they described that two key underpinning philosophies of mental health care exist in NZ. Only the western ‘recovery model’ was explained using Table 1. They will be able to compare the two by showing the indigenous Maori-based model as well as the western model in a diagram or table.

Author Response

- The Introduction carefully describes the people, organizations, statements and guidelines involved in acute mental health care. The authors use diagrams to help experts in other countries better understand the relationship.

We are unclear as to the diagram referred to in the first point.  Diagramming all of the people, organisations, statements and guidelines would be extremely complicated, and we are unsure of how it would aid overseas readers.  If a diagram is deemed essential, we would welcome a bit more direction as to how it might be developed.

We thank this reviewer for their acknowledgement of the novelty of our paper. 

- In the Method section, the current description does not allow other researchers to replicate this study. Did the lead author conduct an analysis of 42 interviews with staff in 4 acute mental health wards and 43 interviews with service users all by only one researcher? The analysis should be performed by at least 3 researchers to ensure objectivity and validity. Did you use the question list or interview guide in the interview? The list or guideline should be submitted as supplementary information.

We have updated the Methods section to allow for ease of replication.  The lead author conducted all interviews (see Line 126 and 127).  The thematic analysis of interviews was conducted by two authors (see Line 146) and a research assistant (she is now acknowledged under acknowledgements as she did not qualify for authorship in other respects).  We have modified this sentence for greater clarity and added initials to clarify the ‘we’.  The questions used to guide these interviews have been added as two files of supplementary information.

- In the Results section, they described that two key underpinning philosophies of mental health care exist in NZ. Only the western ‘recovery model’ was explained using Table 1. They will be able to compare the two by showing the indigenous Maori-based model as well as the western model in a diagram or table.

We have added two diagrams illustrating the Maori models (Figures 1 and 2, page 6 Line 200).  The Maori model is not specific to mental health (includes physical and other dimensions of health) as it is a holistic philosophy regarding wellbeing and it extends beyond the individual to the collective (iwi and hapu).  A comparison side by side of the two (Maori and Western models) would be hard to do without risking being culturally offensive to Maori. 

Reviewer 3 Report

The major limitation of this study was short of originality and novelty. The present study was also deficient in the significance of content, and the quality of presentation should be improved. In addition, the authors should enhance the interest to the readers.

Author Response

The major limitation of this study was short of originality and novelty. The present study was also deficient in the significance of content, and the quality of presentation should be improved. In addition, the authors should enhance the interest to the readers.

This study as noted by Reviewer 2 is highly original and not published in the academic literature.  In fact, this work was funded by a Royal Society of New Zealand grant, the most competitive research fund in New Zealand with the key criteria for funding being novelty and originality of research questions and research design as assessed by a panel of national and international experts. The quality of the presentation was commended by Reviewer 4 but has also been improved through attention to the comments of the three other reviewers and the addition of diagrams and supplementary files.

Reviewer 4 Report

The study highlights a very important connection between mental health and design, especially in conditions of crisis. The presentation of the organisational and legislative context was efficient as well as the description of the methodological design. More details regarding the questionnaire for both staff and users could have been included. It was not clear from the text if both parties were asked in relation to the design of the wards/clinic, their environmental comfort and the provision of safety/privacy from a design perspective. Also, it could have been beneficial (and maybe essential) to include literature related to these spaces coming from environmental psychology or similar studies such as

Connellan, K., Gaardboe, M., Riggs, D., Due, C., Reinschmidt, A. and Mustillo,

(2013), ‘Stressed spaces: Mental health and architecture’, HERD: Health

Environments Research and Design Journal, 6:4, pp. 127–68.

Kronish, N. and Poldma, T. (2013), ‘Exploring peoples’ needs in a psychiatric

ward: Social community, design and the interior environment in a healthcare

setting’, Journal of Design Research, 11:4, pp. 336–50.

Shumaker, S. A. and Reizenstein, J. E. (1982), ‘Environmental factors affecting

inpatient stress in acute care hospitals’, in G. W. Evans (ed.), Environmental

Stress, New York: Cambridge University Press, pp. 179–223.

Muskett, C. (2014), ‘Trauma-informed care in inpatient mental health settings:

A review of the literature’, International Journal of Mental Health Nursing,

23:1, pp. 51–59.

NHS (1999), Safety, Privacy and Dignity in Mental Health Units. Guidance on

Mixed Sex Accommodation for Mental Health Services, London: Department

of Health.

And a more critical perspective of the acute mental health facilities approach. The importance of art, access to nature/nature views, windows (extensive literature on these aspects is available) has not been stressed. Alternative layouts/blueprints could have provided a helpful visualization of table 4 and a comparison between proposed and existing layout/blueprints would have been useful to establish a better picture of the inadequacies of the existing model. There could have been more discussion on alternative models of therapy/approaching mental health from an indigenous perspective. The discussion and conclusions could have been enriched with an elaboration on the proposed ‘holistic approach’ but they were overall satisfactory in terms of answering the research question(s)/purpose of this paper.

Author Response

More details regarding the questionnaire for both staff and users could have been included. It was not clear from the text if both parties were asked in relation to the design of the wards/clinic, their environmental comfort and the provision of safety/privacy from a design perspective.

A copy of the questions used to lead the interviews has been added as supplemental material.

Also, it could have been beneficial (and maybe essential) to include literature related to these spaces coming from environmental psychology or similar studies such as (a list of excellent articles was provided, thank you)

And a more critical perspective of the acute mental health facilities approach. The importance of art, access to nature/nature views, windows (extensive literature on these aspects is available) has not been stressed. Alternative layouts/blueprints could have provided a helpful visualization of table 4 and a comparison between proposed and existing layout/blueprints would have been useful to establish a better picture of the inadequacies of the existing model.

We are aware of much of the literature regarding the architectural details and responses.  This specific paper is the first in a series of papers and in fact we had been writing a more architectural paper when we realised that the purpose of the facility was unclear and required more careful consideration and investigation to understand those interventions and designs that would be most appropriate. 

There could have been more discussion on alternative models of therapy/approaching mental health from an indigenous perspective.

Similarly, a discussion of the alternative models of mental health care therapies from an Indigenous perspective and how they influence design is underway with Maori research colleagues.  This also requires more ‘space’ to be meaningful and is deserving of a paper in its own right.  A cursory commentary would not be culturally appropriate and requires the input of Maori researchers.  We thank this reviewer for their sensitivity in understanding that elaborating on the holistic Indigenous approach would be enriching, but noting that it is currently satisfactory.

The discussion and conclusions could have been enriched with an elaboration on the proposed ‘holistic approach’ but they were overall satisfactory in terms of answering the research question(s)/purpose of this paper.

We concur that the discussion and conclusions could have been enriched by more information with respect to the application of a Maori health model(s).  We are currently working with Maori researchers in this respect and intend on publishing a ‘follow on’ paper.  This paper is the foundation for a series of extended and more detailed publications.

Round 2

Reviewer 2 Report

To Authors:

The manuscript as a whole is that it is well written and clear. This research has novelty as showing the actual challenges in the field that was not well understood so far, and can be useful for developing effective mental health support in the future. 

Author Response

Manuscript ID: ijerph-1038506

Title: Fit for What Purpose?  Developing a conceptual framework for the design of acute mental health facilities

Authors: Gabrielle Jenkin *, Jaqueline McIntosh, Susanna Every-Palmer

Innovations in Architecture for Mental Health

Round 2

Dear editors:

The following is the details of the revisions in the manuscript responding to the reviewers' comments.  

Reviewer 2 Comments:

In the tick box section, this reviewer indicated that the Introduction could be improved.

“The manuscript as a whole is that it is well written and clear. This research has novelty as showing the actual challenges in the field that was not well understood so far, and can be useful for developing effective mental health support in the future.”

Responding to the comments of Reviewer 2, the Editor and Reviewer 3 we have entirely rewritten the Introduction.

Reviewer 3 comments:

The major limitation of this study was short of readers do not know what is the novelty findings after read this manuscript. This study found that the general underpinning philosophy of mental health care in New Zealand was that of recovery, and the CHIME principles of recovery, with some modifications, could be translated into design principles for an architectural brief. However, we can review Bird et al (2014)’s paper and consult with the staff of acute mental health care facilities to reach the same conclusion without executing this study. Please elaborate what is novelty between Bird et al (2014)'s study and this study. In addition, please illustrate the most novelty findings of this study, and compare the similarities and differences with other country/area (e.g. Europe, Asia, Africa, America, Latin America……).

The novelty of our study relates to the bicultural aspects and the architectural implications.  We have modified the title and adjusted the abstract to be clearer about both the novelty and the differences from the Bird study (which did not address either Indigenous health models nor architectural implications).  The title change is hoped to remove any similarity.

The wordcount does not permit a comparison of similarities and differences with other countries and continents such as Europe, Asia, Africa, Latin America etc.  We have invited overseas researchers to submit research that will address this in the necessary detail.

Reviewer 3 Report

The major limitation of this study was short of readers do not know what is the novelty findings after read this manuscript. This study found that the general underpinning philosophy of mental health care in New Zealand was that of recovery, and the CHIME principles of recovery, with some modifications, could be translated into design principles for an architectural brief. However, we can review Bird et al (2014)’s paper and consult with the staff of acute mental health care facilities to reach the same conclusion without executing this study. Please elaborate what is novelty between Bird et al (2014)'s study and this study. In addition, please illustrate the most novelty findings of this study, and compare the similarities and differences with other country/area (e.g. Europe, Asia, Africa, America, Latin America……).

Author Response

(The authors gave the same response as above.)

Round 2

Dear editors:

The following is the details of the revisions in the manuscript responding to the reviewers' comments.  

Reviewer 2 Comments:

In the tick box section, this reviewer indicated that the Introduction could be improved.

“The manuscript as a whole is that it is well written and clear. This research has novelty as showing the actual challenges in the field that was not well understood so far, and can be useful for developing effective mental health support in the future.”

Responding to the comments of Reviewer 2, the Editor and Reviewer 3 we have entirely rewritten the Introduction.

Reviewer 3 comments:

The major limitation of this study was short of readers do not know what is the novelty findings after read this manuscript. This study found that the general underpinning philosophy of mental health care in New Zealand was that of recovery, and the CHIME principles of recovery, with some modifications, could be translated into design principles for an architectural brief. However, we can review Bird et al (2014)’s paper and consult with the staff of acute mental health care facilities to reach the same conclusion without executing this study. Please elaborate what is novelty between Bird et al (2014)'s study and this study. In addition, please illustrate the most novelty findings of this study, and compare the similarities and differences with other country/area (e.g. Europe, Asia, Africa, America, Latin America……).

The novelty of our study relates to the bicultural aspects and the architectural implications.  We have modified the title and adjusted the abstract to be clearer about both the novelty and the differences from the Bird study (which did not address either Indigenous health models nor architectural implications).  The title change is hoped to remove any similarity.

The wordcount does not permit a comparison of similarities and differences with other countries and continents such as Europe, Asia, Africa, Latin America etc.  We have invited overseas researchers to submit research that will address this in the necessary detail.